# Facilitating stakeholder engagement in early stage translational research

Amy M. LeClair[1,2,3]*, Virginia Kotzias[4], Jonathan Garlick[2], Allison M. Cole[5], Simona C. Kwon[6], Alexandra Lightfoot[7,8], Thomas W. Concannon[2,3,4,9]

1 Tufts Medical Center, Boston, MA, United States of America, 2 Tufts University School of Medicine, Boston, MA, United States of America, 3 Tufts Clinical and Translational Sciences Institute, Boston, MA, United States of America, 4 RAND Corporation, Arlington, VA, United States of America, 5 Institute of Translational Health Sciences, University of Washington School of Medicine, Seattle, WA, United States of America, 6 Department of Population Health, School of Medicine, New York University, New York, NY, United States of America, 7 Gillings School of Global Public Health, University of North Carolina at Chapel Hill, Chapel Hill, NC, United States of America, 8 North Carolina Translational and Clinical Sciences Institute, University of North Carolina at Chapel Hill, Chapel Hill, NC, United States of America, 9 RAND Corporation, Boston, MA, United States of America

* alecair@tuftsmedicalcenter.org

## Abstract

### Introduction

Stakeholder engagement can play an important role in increasing public trust and the understanding of scientific research and its impact. Frameworks for stakeholder identification exist, but these frameworks may not apply well to basic science and early stage translational research.

### Methods

Four Clinical and Translational Science Award (CTSA) hubs led six focus groups and two semi-structured interviews using a semi-structured discussion guide to learn from basic science researchers about stakeholder engagement in their work. The 24 participants represented fourteen clinical and academic disciplines.

### Results

Early stage translational researchers reported engagement with a broad array of stakeholders. Those whose research has a clinical focus reported working with a more diverse range of stakeholders than those whose work did not. Common barriers to stakeholder engagement were grouped into three major themes: a poor definition of concepts, absence of guidance, and limited resources.

### Discussion

The National Center for Advancing Translational Sciences (NCATS), the consortium of CTSAs, and the individual CTSA "hubs" are three actors that can help early stage translational researchers develop shared terms of reference, build the necessary skills, and

**Data Availability Statement:** Data cannot be shared because participants provided verbal consent to participate after being explicitly told that the data would not be shared beyond the research team. This study was deemed exempt by Tufts

Health Sciences Institutional Review Board, therefore a written informed consent was not used. However, participants were given an information sheet and verbal consent was obtained prior to beginning the focus groups/interviews. During this process, participants were explicitly told that their data would not be shared outside of the research team, and that only de-identified excerpts of the transcripts would be used in publication and other forms of dissemination. Furthermore, the complete data set from all four sites was only available to three members of the team for coding. We have included the codebook that was developed and used for analysis.

**Funding:** This study was funded by a grant to Tufts University (UL1 TR001062) which provided support to AML, TC, JG; a grant to the University of Washington (UL1 TR000423) which provided support to AC; a grant to New York University (UL1 TR001445) which provided support to SK; and a grant to the University of North Carolina, Chapel Hill, which provided support t AL (UL1 TR001111).

**Competing interests:** The authors have declared that no competing interests exist.

assemble the appropriate resources for engaging stakeholders in Clinical and Translational Research. Getting this right will involve a coordinated push by all three entities.

## Introduction

Engaging stakeholders early in the translational spectrum could help to advance public trust and understanding of science and the impact of scientific research on human health. The National Center for Advancing Translational Science (NCATS) and other research funders have made substantial investments in developing approaches and resources to support engagement in translational research [1]. Established frameworks for stakeholder identification and involvement in research exist for clinical and outcomes research [2–4], but it is reasonable to question the extent to which these apply to early stage translational science.

Four Clinical Translational Science Award (CTSA) hubs launched a collaboration to explore how to engage with stakeholders in the setting of T0 (basic biomedical research) and T1 (translation to humans) research. The "T's" in the translation research spectrum represent the transitions between the phases of research [5]. We set out to answer three research questions: (1) Who are the stakeholders in early stage translational science? (2) How can CTSA institutions and researchers improve stakeholder engagement in early stage research? (3) What are the barriers and facilitators to engaging stakeholders in early stage translational research?

Clinical and translational research (CTR) is the process of turning scientific observations into interventions that improve and enhance the health and well-being of individuals and populations [6]; basic science is research that addresses foundational questions in the earliest stages of translation. We define a *stakeholder* as an individual or group who is responsible for or affected by health- and healthcare-related decisions that can be informed by research evidence. We define *engagement* as a bi-directional relationship between the stakeholder and researcher that results in informed decision-making about the selection, conduct, and application of research findings [2].

## Methods

This study was approved by the Tufts University/ Tufts Medical Center Health Sciences IRB. IRB approval #12224—This study was deemed exempt. The Clinical and Translational Science Institute at Tufts University (Tufts CTSI) convened its own and three additional CTSA hubs–the Institute for Translational Health Sciences at the University of Washington (ITHS), the Clinical and Translational Science Institute at New York University Langone Health (NYU CTSI), and the Translational and Clinical Sciences Institute at the University of North Carolina, Chapel Hill (NC TRaCS)–to investigate the views of early stage translational science researchers on the involvement of stakeholders in their work. Data were collected via focus groups and semi-structured interviews. Each CTSA recruited discussants from lists of T0 and T1 researchers who had accessed resources in their own hubs.

A focus group discussion guide was developed from a simple logic that ties research studies directly to decision problems faced by stakeholders: decisions made by stakeholders can be informed by evidence; the need for this evidence can be formed into a topic and question; and research can be developed to address this topic and question. The guide posed three key broad questions: (1) how is your work used in other applications; (2) who uses your work in other applications; (3) who is affected by your work as it is used in other applications? Probes included

assessing the barriers and facilitators to achieving the ideal answer to each question. Introductory material, probes, and instructions to the interviewer were included (S1 Appendix).

The guide was pilot tested with two researchers at Tufts CTSI and revised to improve the clarity of the questions. Focus groups were held for one hour, interviews lasted approximately 30 minutes, and both were audio recorded. Focus groups and interviews at non-Tufts sites were administered via WebEx and facilitated by investigators at Tufts with extensive experience in qualitative data collection to ensure consistency across sites. Audio recordings were transcribed verbatim and deleted after transcripts were de-identified. De-identified transcripts were coded using Dedoose™. A codebook was developed deductively from the discussion protocol based on previous literature on stakeholder engagement [7,8]. Two coders (AL and VK) reviewed each transcript independently and added emergent themes identified using a modified grounded theory approach [7, 9]. Once they finalized the codebook, the coders reanalyzed transcripts and used a comparison and consensus approach to resolve any discrepancies [10]. After coding was complete, we continued to iteratively group categories of codes into the broader categories of "barriers" and "facilitators" until the major themes discussed below crystalized as unique but related concepts [7].

## Results

We convened six focus groups (Tufts CTSI = 3; ITHS = 2; NYU CTSI = 1), and, at one site (NC TRaCS) where convening a focus group was not feasible, we conducted two interviews using the same discussion guide. The focus groups ranged in size from two to five participants. Ultimately, we held eight conversations with 24 individuals representing a range of clinical, methodological, career stage characteristics, and previous experience with stakeholder engagement in their work (Table 1). Participants' understandings of stakeholder engagement, their views on barriers that stand in the way of engaging stakeholders, and their recommendations for introducing new facilitators to support engagement work are described below.

### Stakeholder engagement in early stage translational science

All early stage translational researchers reported some level of engagement, and many had engaged with a broad array of stakeholders, but not all engaged with the same groups of

**Table 1. Participant demographics.**

| Characteristic | % (N) |
|---|---|
| Female | 42% (10) |
| Academic rank | |
| Assistant | 42% (10) |
| Associate | 25% (6) |
| Full | 33% (8) |
| MD | 33% (8) |
| Departments/Disciplines represented: | |
| Anesthesiology | Neuroscience |
| Cardiology | Obstetrics/Gynecology |
| Chemistry | Pediatrics |
| Gastroenterology | Pharmacology |
| Immunology | Pulmonary |
| Medical Oncology | Rheumatology |
| Molecular Biology | Speech Language Pathology |

stakeholders as their colleagues. Researchers working on specific diseases or clinical conditions had engaged with federal and local government agencies (the Centers for Disease Control, the Department of Defense, the Department of Agriculture, local branches of Health and Human Services), international government bodies (a foreign department of health, the World Health Organization), private industry, patients and advocacy groups, and many other types of stakeholders. One researcher, whose focus is a parasite common in cows, talked about working with farmers in the country where she conducted her work. The researcher's passion was children affected by the disease, but the financial impact on the local agricultural community produced more engagement, highlighting the ways in which factors external to the research can shape engagement. Those doing clinically agnostic work, such as the mechanisms of cell death, were more likely to have engaged only with other researchers as stakeholders in their work. As one chemist stated: "My fondest desire for an end user is another basic scientist."

Working with practitioners to define potential clinical applications of early stage research was important to some researchers. A senior basic scientist said, "It's very important to have the clinicians on board early," because it helps ensure the work will be of clinical relevance. Another said, "We regularly have medical doctors as trainees in my laboratory to make sure the stuff we do is of medical interest."

## Barriers and facilitators to engaging stakeholders in early stage research

Participants described a wide range of barriers to engaging stakeholders. Barriers were grouped iteratively until three major themes emerged. The three themes and related recommendations to address them–the facilitators–are described below under the following themes with titles drawn directly from transcripts.

**Theme 1. "Poor definitions" and "Translation is in the eye of the beholder".** The terms of reference we use in CTR emerged as a barrier. Who qualifies as a stakeholder, what constitutes engagement, and how one understands translation were all points of confusion and contention. While definitions exist, there appeared to be a lack of shared understanding among many participants.

Whether research is deemed translational may be a matter of definition. Some participants felt uneasy with the general concept of "translational science" and viewed it as at odds with the fundamentally incremental nature of science. Some worried that a "rush" to translation could lead to skipping steps or avoiding interesting avenues of research that could lead to new discovery. One researcher said,

*I struggle with the definition of what's 'translational research' and who our work is supposed to affect. I think by using the word 'translational,' you tie it directly to patients, yet work in disease models is the fundamental first step to this. And so I guess I'm just somewhat uncomfortable even with the term 'translational research' in this whole need to directly tie everything we do immediately to some end outcome.*

Another researcher suggested a different emphasis is needed to address research in T0 and T1 settings:

*We really have a push at NIH to be translational and I don't think that people really understand what that means. . .It doesn't mean that you don't do basic science or that everything has to be a model of some disease. . .[The] frustration is [with] the idea that every scientific work has to be transformational and not incremental.*

Participants also shed light on what defines a stakeholder and which stakeholders are important in their work. No one disagreed with the definition of stakeholders that we put forward, the same one that is presented in frameworks for identifying stakeholders in T2 through T4 CTR(2)–namely that stakeholders are those who make decisions with evidence or are affected by the decisions that are made with evidence. However, several participants pointed out that this previous framework's call to scan the "7Ps" (patients, providers, payers, purchasers, product makers, policy makers, and principal investigators) for relevant decision-makers obscures a necessary emphasis on involving one "P" in particular: the principal investigators (researchers, research entities, and research funders) doing similar basic science work in other disciplines. This special emphasis amounts to a call for multi-disciplinary research to broaden the evidence base before basic discoveries begin translation to humans: "There's a lot of stuff in the 'Stage 1's' of translational research that requires a lot of intensive investigations for a lot of potential applications."

**Recommendations to address Theme 1.** Researchers expressed a need a for better direction from NCATS and their CTSA hubs about the terms of reference surrounding engagement work in early stage translational science: definitions of translation need to allow for discovery research to proceed without being linked strictly to a clinical, intervention, or product pathway; identification of relevant stakeholders should start with an emphasis on researchers and research groups representing multiple disciplines; and guidance on identifying relevant stakeholders should shift to the "7Ps" framework only as clinical applications, interventions, and product pathways emerge from discovery.

Building the case for using T0 and T1 evidence in other applications often involves collaboration with a cohort of other principal investigators, often a multi-disciplinary team, before the work can progress from early stage to latter stage research. As careers progress, the more distal end users of T0 and T1 work may become apparent, as the investigator sees their work wending its way into T2 and T3 settings. Here, the end users may be the full complement of stakeholders: "We all want to feel that what we're doing is important and will one day lead to something, whether it's directly from what we conceive or it might be a step in the process. . .That maybe we don't have the magic, but what magic we develop will be able to be part of the step, then, to the final process."

**Theme 2: "No instructions" and "Tell us what to do, in what order".** The second theme that emerged was a skills barrier: This is the skills issue: absence of guidance, training, mentorship and skills.

For early career investigators the absence of training on practical approaches to stakeholder engagement is a barrier. For instance, a junior investigator said he did not need to be sold on the concept of stakeholder engagement in translational science; he bought it, "hook, line, and sinker." What he lacked was instructions about which stakeholders to engage and when in the process to engage them:

> *I don't even know what that pathway is. There's not really a trail that says, 'Do this, then do this, then do this. Speak with these people along the way.' I don't know what the path is or who the path is through, I just know very broadly where it needs to go.*

Another echoed this comment: "Everyone wants to translate something, right? It's a buzz word. Everyone wants to do that thing or say they've done that thing, but the *'hows'* are not as well defined for at least people in my space." Similarly, some participants talked about a lack of training in the mechanics of translating bench discoveries into clinical applications, interventions, or products. For instance, one participant focused on administrative hurdles they discovered when working with industry on patents and licensing: "[I] feel as a junior faculty that

I don't necessarily have the commercial training nor the time to learn this skill as I would perhaps after tenure."

Some researchers reported lacking the skills necessary to communicate scientific ideas to lay audiences in simple but meaningful ways. Good science communication skills were seen as crucial: "No matter how good their ideas, you have to be able to explain it to whoever it is you're talking to if you want them to pay attention;" but lacking: "Scientists aren't exactly the best spokespeople for their own work much of the time."

Lack of mentoring was also challenging for junior faculty. Variations in the culture of the discipline, department, or institution could also lead to researchers feeling isolated or silo-ed. "I feel a little isolated where I am. I don't necessarily have a mentor to help guide me in a lot of the decision processes that I'm making." The clinician-researcher role was seen as better established in some fields (cardiology) than others (gynecology), although this may vary between institutions, and this impacted both the availability of protected research time for clinicians and the availability of mentors. In non-clinical fields (chemistry), the proximity (or lack thereof) of the department to the healthcare setting could also create a barrier:

> *Very few of my colleagues are involved with anything that is beyond foundational basic research. And even fewer are involved in anything related to healthcare. I don't necessarily have a mentor to help guide me in a lot of the decision processes that I'm making. That just adds to a feeling of isolation and flying by the seat of pants in not being savvy regarding who I contact about collaborations or grants because there's no one telling me, 'That's a dumb idea, that's a good idea.'*

**Recommendations to address Theme 2.** Several recommendations were made to address the barriers described by skills-related barriers. These include: training that folds engagement work into researcher education, creation of practical "how to" guides on engagement work, and more opportunities to use mentoring in CTR, including on how to identify and become engaged with potential en-users of the work.

> *You have at least two junior faculty members [here] that are begging for this sort of pathway forward. Tell us what to do, in what order, how to contact people, when's the right time to engage this stakeholder and this stakeholder and this stakeholder.*

CTSA hub investments in mentorship programs were identified as a way to overcome these barriers. After describing the absence of a clear clinician-research path in her field and institution, one participant said that the mentorship she encountered through her institution's CTSA from colleagues outside her field had greatly benefited her: "One thing that [the CTSA] has given me that's been a real boon is the mentorship. . .without that, I couldn't have done it."

**Theme 3: "Competing demands on resources" and "lack of resources".** Material resources to support research were viewed by our discussion participants as scarce, may have strings, and difficult to come by. All of this puts pressure on researchers to choose among the many competing demands of research, and stakeholder engagement can fall off the radar without funds that are directed specifically to that purpose, or without funding criteria that call for it.

Having protected time for research was one of the most basic challenges. Junior faculty without established research portfolios felt this acutely:

> *My chair is very morally supportive but nobody gives you startup money or protected time. So I started out, after my Fellowship, doing 100% clinical, and I had to rearrange my time in*

*order to have a little bit of research time that wasn't 9:00 at night. That was very stressful, because I have the research knowledge, but I didn't have the time or energy to do it. And, slowly, I've crawled my way to having more protected time through luck and opportunity and meeting people–enough protected time that I was able to apply for this award [institutional K], but it's taken me four and a half years.*

A tight funding climate was described by all participants as a barrier to adding stakeholder engagement to their list of priorities, but tight funding presents a unique challenge for junior researchers. Some participants described being unable to find a "home" in the National Institutes of Health (NIH), because their research interests do not line up with individual Institutes and Centers of NIH: "I don't really have an institute in the NIH where I can apply very easily." An early career researcher said, "The mentality of the tenure track is just go, go, go, go, go, which is great. But sometimes I just wish I had the time to just sit back, think logically of the next step."

Some participants described funding challenges in another context: the misalignment of industry profit motives and the patient or public health goals of researchers. Returning to an earlier example of a research who wanted to prioritize the health of humans over livestock, they state: "We have licensed an antibody that we developed to use for treatment of [this bacteria] . . . I've sort of hit a wall against finding somebody who is interested in, in developing the technology or marketing it for use in developing countries because there's no money in it."

**Recommendations to address Theme 3.** To successfully engage stakeholders in the earliest stages of translational research, researchers need material resources that are specifically targeted to that purpose. CTSAs are in a unique position to deliver many of these material resources, and participants recognized this:

*I think that the [CTSA] is an important space that values research and, maybe, could do some more pushing or open up some more opportunities for people who are in the clinical field that really would like to pursue research.*

Participants had the fewest suggestions for how to address funding barriers. Everyone recognized it as a problem, but no one purported to have the solution. It may be this is where institutions, such as NCATS, need to take the lead in helping to create structures that support protected time for research and access to funding which prioritizes stakeholder engagement especially for researchers in the earliest stages of CTR.

## Discussion

In mathematics, a lemma is known as a "helping theorem," a rule that may be used to develop or support some result. We use the term "three lemmas" to call attention to three helping theorems by which barriers described in this article may be eliminated, and we describe the organizations who may take charge of this effort: the National Center for Advancing Translational Sciences (NCATS), the consortium of CTSAs, and the individual CTSA "hubs" are three organizational entities that can help T0 and T1 researchers develop shared terms of reference, build the necessary skills, and assemble the appropriate resources for engaging stakeholders in basic science research. Getting this right will involve a coordinated push by all three entities; therefore, this is a triple helping theorem with three entities playing important roles.

There is tremendous variation in the level of stakeholder engagement among researchers in early stage (T0-T1) translational science. Participants identified a number of barriers to engaging stakeholders. We heard also, in these conversations, about the facilitators–the resources

that help to remove those barriers. Below we summarize those barriers and make recommendations for how to address them.

## Addressing definitions

There seems to be a problem with the operational definition of translational research. We have identified numerous definitions of translational research, and while there is overlap, there is not total consensus. Some researchers we spoke with are conflating "translational" with "patient centered," and while we might agree that good translational research *is* patient-centered, focusing *solely* on patients as end-users comes at the exclusion of multiple other stakeholder groups and may represent a cognitive barrier to stakeholder engagement. CTSAs can help with emphasizing the *spectrum* of translational research and the full range of stakeholders that can be engaged along that spectrum. The multi—nodal, concentric rings presented in the Translational Research Framework put forth by the National Institute of Environmental Health Sciences (NIEHS) is a useful diagram for visually representing the complexity and possibilities of CTR. Translational can occur by moving along one's own circle, to a different discipline (node), for example. Reaching the outer circle of population health can be the ultimate goal, but there can be plenty of movement within the other circles. Translational science can be a marathon or a relay–every researcher or research team does not have to go the full distance on their own.

## Instruction manual

Even for those for whom the definition of translational research is not an impediment, the process can seem impenetrable. The *what* is not the charge, but the *how* is. The good news is that frameworks exist [2–4, 11]. We need to do a better job of disseminating these guidelines and actively targeting researchers in the earliest stages of CTR. In the Clinical and Translational Science graduate program at Tufts CTSI, graduate students are required to have a stakeholder on their thesis committee. The Clinical and Translational Science Institute at New York University's Langone Health prioritizes funding pilot studies that demonstrate strong stakeholder engagement. At the NC TraCS Institute at the University of North Carolina, the Community and Stakeholder Engagement (CaSE) unit has developed a range of tools and trainings to promote stakeholder engagement, build researcher capacity and provide technical assistance to investigators across the translational spectrum [12].

## Focusing the CTSA hubs to deliver material support for engagement work in basic science research

When it comes to resources, the good news is that there already exists a mechanism designed specifically to address the issue of providing resources to researchers engaging in translational research. This is, in fact, the very mandate of NCATS. While CTSAs cannot replenish the coffers at NIH, they can provide resources–pilot grants, grant writing workshops, mentorship–that can help support researchers in their pursuit of funding and achieving the research once it is funded.

Of the recommendations we have made here, 1 (definitions) and 2 (instructions) are necessary, but not sufficient. Even with crystal clear definitions and guidelines, it is ultimately 3, the resources, that make the difference. These issues are particularly challenging for young researchers because they are the most strapped for resources. While an initial challenge may be to convince people of the value of translational research, this may require less of a paradigm shift for early career investigators and researchers. CTSAs should embrace and encourage their enthusiasm by supporting them via focused training and resources.

CTSAs are charged with facilitating the translation of science. Stakeholder engagement is a key component of this mission. Paying special attention to the unique needs of research working in at the earliest stages of the translational spectrum and providing them with resources to overcome the barriers they encounter could facilitate the advancement of the science itself as well as public trust and understanding of science and the impact of scientific research on human health.

## Supporting information

**S1 Table. Codebook.**
(DOCX)

**S1 Appendix. Discussion guide.**
(DOCX)

## Acknowledgments

The authors would like to thank the following individuals for their assistance with this project: Lisa Quarles, BA, Training Coordinator and Community Engagement Specialist for the North Carolina Translational and Clinical Sciences Institute, home of the CTSA at the University of North Carolina at Chapel Hill; Smiti Kapadia Nadkarni, MPH, Program Manager for Community Engagement and Population Health Research Program (CEPHR), NYU-H+H Clinical & Translational Science Institute.

## Author Contributions

**Conceptualization:** Amy M. LeClair, Thomas W. Concannon.

**Data curation:** Amy M. LeClair, Virginia Kotzias, Thomas W. Concannon.

**Formal analysis:** Amy M. LeClair, Virginia Kotzias.

**Funding acquisition:** Thomas W. Concannon.

**Investigation:** Amy M. LeClair, Allison M. Cole, Simona C. Kwon, Alexandra Lightfoot, Thomas W. Concannon.

**Methodology:** Amy M. LeClair, Allison M. Cole, Simona C. Kwon, Alexandra Lightfoot, Thomas W. Concannon.

**Project administration:** Amy M. LeClair, Virginia Kotzias, Allison M. Cole, Simona C. Kwon, Alexandra Lightfoot, Thomas W. Concannon.

**Resources:** Amy M. LeClair, Allison M. Cole, Simona C. Kwon, Alexandra Lightfoot, Thomas W. Concannon.

**Writing – original draft:** Amy M. LeClair, Virginia Kotzias, Thomas W. Concannon.

**Writing – review & editing:** Jonathan Garlick, Allison M. Cole, Simona C. Kwon, Alexandra Lightfoot.

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
