## [Decision Letter · Decision Letter 0]

26 Feb 2020

PONE-D-19-35056

Facilitating Stakeholder Engagement in Early Stage Translational Research

PLOS ONE

Dear Dr LeClair,

Thank you for submitting your manuscript to PLOS ONE. After careful consideration, we feel that it has merit but does not fully meet PLOS ONE’s publication criteria as it currently stands. Therefore, we invite you to submit a revised version of the manuscript that addresses the points raised during the review process.

ACADEMIC EDITOR: 

I found the study interesting. Please make overt the framework used to inform the maintainence of rigour throughout the qualitative data collection and analysis. Also further detail about how the themes emerged is required. Please disregard the reviewer 2 comment about quantitative research as this does not apply to your paper.

We would appreciate receiving your revised manuscript by 20/05/2020. To enhance the reproducibility of your results, we recommend that if applicable you deposit your laboratory protocols in protocols.io, where a protocol can be assigned its own identifier (DOI) such that it can be cited independently in the future. For instructions see: http://journals.plos.org/plosone/s/submission-guidelines#loc-laboratory-protocols

We look forward to receiving your revised manuscript.

Kind regards,

Karen-Leigh Edward

Academic Editor

PLOS ONE

Journal Requirements:

We note that one or more of the authors are employed by a commercial company: RAND Corporation.

Reviewers' comments:

Reviewer's Responses to Questions

**Comments to the Author**

1. Is the manuscript technically sound, and do the data support the conclusions?

Reviewer #1: Yes

Reviewer #2: No

2. Has the statistical analysis been performed appropriately and rigorously? 

Reviewer #1: N/A

Reviewer #2: N/A

3. Have the authors made all data underlying the findings in their manuscript fully available?

Reviewer #1: Yes

Reviewer #2: No

4. Is the manuscript presented in an intelligible fashion and written in standard English?

Reviewer #1: Yes

Reviewer #2: Yes

5. Review Comments to the Author

Reviewer #1: The terms CTSA, T0, T1, T2, should be explained the first time each is used in the manuscript.

On page 2 where it says 'are three actors can help" should read "are three factors".

On page 11 where it says "by the set of pants" should probably read "by seat of pants"

On page 14 "we describe the actors who may take charge". Actors should be changed to a more scientific term such as participants or entities.

I thought the parts "Addressing definitions" and "Instruction manual" were both accurate and insightful.

The section "Focusing on CTSA hubs...research", clearly outlines hoe limited access to funding affects all aspects of the scientific research process.

Reviewer #2: The major concern about the current study is qualitative and the findings were from interviews and subjective. The topic is interesting, if conduct the study by design a survey with questionnaire and collect the data from the identified stakeholders, and then analyze the data, according to the response rate, the reliability and validity would allow the study more informative and objective.

6. PLOS authors have the option to publish the peer review history of their article (what does this mean?). If published, this will include your full peer review and any attached files.

Reviewer #1: Yes: Frank Abbruscato

Reviewer #2: No

---

## [Author Response · Author response to Decision Letter 0]

29 May 2020

Dear Karen-Leigh Edward and reviewers,

Thank you for the thoughtful reviews and opportunity to respond. We have organized our responses below:

1. Style requirements – We have reviewed our files to ensure they meet PLOS ONE’s style requirements. 

2. Data availability – While we understand data sharing has become more common for qualitative data, we are new to the concept for the sharing of qualitative data. We realize, in retrospect, we should have addressed this when submitting the manuscript, and the lead author – Dr. Amy LeClair – consulted with the Qualitative Data Repository (QDR) group for guidance. What we should have said in the original submission is that the data cannot be shared. We appreciate the opportunity to learn from this experience, both for the submission of future manuscripts and for the design of qualitative studies going forward.

a. This study was deemed exempt by Tufts Health Sciences Institutional Review Board, therefore a written informed consent was not used. However, participants were given an information sheet and verbal consent was obtained prior to beginning the focus groups/interviews. During this process, participants were explicitly told that their data would not be shared outside of the research team, and that only de-identified excerpts of the transcripts would be used in publication and other forms of dissemination. Furthermore, the complete data set from all four sites was only available to three members of the team for coding. 

3. ORCID ID – The corresponding author now has an ORCID ID, and it has been validated in Editorial Manager.

4. Competing Interests – The RAND Corporation is not a commercial company. RAND has the same nonprofit status as the academic institutions and medical centers the other authors’ are affiliated with. It would not be appropriate to declare this as a competing interest and is not the policy of RAND or its researchers and staff. We have, therefore, not updated any of our Funding Statements or Competing Interests Statements

5. Supporting Information – We have including captions for our supporting Information files.

6. Reviewers’ Comments:

Comment Response

Academic Editor: “Please made overt the framework used to inform the maintenance of rigour throughout the qualitative data collection and analysis. Also further detail about how the themes emerged is required.” We have made several edits to the 3rd paragraph of the methods section to clarify the framework that guided our data collection and analysis, including adding citations to the literature that guided our methods. We hope this clarifies the methodology. 

Reviewer #1: The terms CTSA, T0, T1, T2, should be explained the first time each is used in the manuscript. We have made sure all acronyms and abbreviations are spelled out in their first use and have also added a citation for our definitions of T0 and T1 in the 2nd paragraph of the article.

On page 2 where it says 'are three actors can help" should read "are three factors". We have changed the word “actors” to “organizational entities” to more clarify t our meaning. 

On page 11 where it says "by the set of pants" should probably read "by seat of pants" Yes – thank you for picking up that type. We have made the correction.

On page 14 "we describe the actors who may take charge". Actors should be changed to a more scientific term such as participants or entities. We have changed “actors” to “organizations.”

I thought the parts "Addressing definitions" and "Instruction manual" were both accurate and insightful.

The section "Focusing on CTSA hubs...research", clearly outlines hoe limited access to funding affects all aspects of the scientific research process. Thank you.

Reviewer #2: The major concern about the current study is qualitative and the findings were from interviews and subjective. The topic is interesting, if conduct the study by design a survey with questionnaire and collect the data from the identified stakeholders, and then analyze the data, according to the response rate, the reliability and validity would allow the study more informative and objective. Per the Academic Editor – “Please disregard the reviewer 2 comment about quantitative research as this does not apply to your paper.”

---

## [Editor Report · Decision Letter 1]

16 Jun 2020

Facilitating Stakeholder Engagement in Early Stage Translational Research

PONE-D-19-35056R1

Dear Dr. LeClair

We’re pleased to inform you that your manuscript has been judged scientifically suitable for publication and will be formally accepted for publication once it meets all outstanding technical requirements.

Kind regards,

Karen-Leigh Edward

Academic Editor

PLOS ONE
---

## [Editor Report · Acceptance letter]

22 Jun 2020

PONE-D-19-35056R1 

Facilitating Stakeholder Engagement in Early Stage Translational Research 

Dear Dr. LeClair:

I'm pleased to inform you that your manuscript has been deemed suitable for publication in PLOS ONE. Congratulations! Your manuscript is now with our production department. 

Kind regards, 

on behalf of

Professor Karen-Leigh Edward 

Academic Editor

PLOS ONE